# Sustainability of Diets Consumed by UK Adolescents and Associations Between Diet Sustainability and Meeting Nutritional Requirements

**DOI:** 10.3390/nu17132140

**Published:** 2025-06-27

**Authors:** Ayesha Ashraf, Marie Murphy, Rhona Duff, Peymane Adab, Miranda Pallan

**Affiliations:** 1Department of Applied Health Sciences, School of Health Sciences, College of Medicine and Health, University of Birmingham, Birmingham B15 2TT, UK; afa824@alumni.bham.ac.uk (A.A.); p.adab@bham.ac.uk (P.A.); m.j.pallan@bham.ac.uk (M.P.); 2Leeds Teaching Hospitals NHS Trust, Leeds LS9 7TF, UK

**Keywords:** adolescents, diet, dietary recommendations, sustainability, planetary health diet

## Abstract

**Background/Objectives**: The EAT-Lancet Commission proposed a healthy and sustainable ‘planetary health diet’ (PHD) based on the nutritional needs of adults, but recommended for all population groups over the age of two years. This study investigated the extent to which UK adolescent diets meet these recommendations, and the association between meeting recommendations and nutritional intake. **Methods**: Dietary intake data collected from students aged 11–15 years between 2019 and 2022 as part of the Food provision, cUlture and Environment in secondary schooLs (FUEL) study was used. Specifically, 24 h intakes of food groups and key nutrients were summarised for each participant, and micronutrient intake was expressed as a percentage of the reference nutrient intake (RNI). The proportion of participants meeting each PHD recommendation was calculated. A PHD index score was developed to measure the overall adherence to recommendations. Multilevel linear regression models were fitted to explore the association between the PHD score and intake of each nutrient. **Results**: No participants fully met the recommendations. The mean PHD score was 14.2 (3.3) out of 39. Adherence was low for the emphasised foods, and lowest for nuts (1.3%) and unsaturated oils (0.2%). Higher PHD index scores were associated with lower intakes of all micronutrients, with the largest association for vitamin B12 [regression coefficient: −12.9% (95% CI: −16.3, −9.4) of RNI] and the smallest for vitamin D [regression coefficient: −0.4 (95% CI: −0.7, −0.2) of RNI]. **Conclusions**: Substantial dietary changes are needed for this population group to meet the PHD guidelines. Further research should focus on why these recommendations may lead to the inadequate intake of some nutrients in this age group. We propose that the PHD guidance needs to be reviewed and potentially adapted for this specific population, with consideration of the typical dietary behaviours of this age group.

## 1. Introduction

The current food systems contribute to the climate crisis. Food production is responsible for a third of human-caused greenhouse gas emissions as a result of the methane produced by ruminant livestock, nitrogen-based fertilisers, and land clearance [1,2]. There is an urgent need for patterns of food consumption to change to reduce the impacts of food systems on the climate crisis.

The dietary changes required to address the climate crisis have the potential to impact health. Currently, the majority of individuals’ diets in the UK are suboptimal when compared to government dietary recommendations, particularly amongst adolescents. According to the National Diet and Nutrition Survey (NDNS), only 11% of individuals aged 11–18 meet the recommendation to consume five different fruits and vegetables a day (termed ‘5-A-Day’), 8% consume within the recommended intake of free sugars, and 4% consume sufficient fibre [3]. Similar dietary patterns are observed across Europe [4] and the US [5]. Differences have been observed by sex, for example, more girls meet the recommendations for saturated fats [3], and by ethnic group, with white British adults more likely to meet the ‘5-A-Day’ recommendation [6].

The EAT-Lancet Commission aims to actively influence individual dietary choices to address both these issues. The Commission has developed the ‘planetary health diet’ (PHD), which aims to allow for a growing global population to be fed a healthy diet while maintaining sustainable food production [7]. PHD recommendations are provided for intakes of various food groups in grams per day, presented as a single target value alongside a minimum and maximum recommendation. For example, it is recommended that 13 g of egg are consumed a day, with a minimum intake of 0 g and maximum of 25 g. The PHD is predominantly plant-based and emphasises fruit, vegetables, legumes, nuts, and wholegrains.

Although the guidelines are aimed at the population aged two years and older, the modelling to determine the nutritional adequacy of the diet is based on an average adult aged 30 years consuming 2500 kcal/day [7]. It is therefore unclear whether the PHD provides adequate intakes of nutrients for children and adolescents. Adolescents have differing requirements for some nutrients compared to adults. For example, higher calcium levels are required for skeletal growth, and high iron levels are required for increasing muscle mass and the expansion of blood volume [8,9].

Plant-based diets encompass a variety of eating patterns, but have generally been shown to be beneficial for the intakes of some nutrients such as fibre, while being associated with inadequate intakes of some micronutrients, such as vitamin B12 [10]. Conversely, saturated fats (which increase the risk of cardiovascular disease) are found in high concentrations in red meat and many dairy foods [11]. Red and processed meats increase the risk of colorectal cancer [12], but provide complete protein intake and more bioavailable sources of iron [13,14].

In the UK, dietary reference values (DRVs) are used to estimate nutritional requirements in groups of healthy people [15]. Reference nutrient intakes (RNIs) outline the DRVs for micronutrients (vitamins and minerals), and refer to the amount of a nutrient that is enough to ensure that the needs of 97.5% of a population group are met. These vary based on age and sex. On average, adolescents in the UK are meeting 70% of the RNIs for iron, 71% for calcium, and 16% for vitamin D [3].

Studies on adherence to the PHD in adolescent and young adult populations have shown inconsistent results in the US and Germany. In Minnesota, Ludwig-Borycz et al. reported a low average consumption of wholegrains, legumes, and fish [16]. In Germany, Vallejo et al. reported high adherence to recommendations for these food groups [17]. Both studies reported the low consumption of nuts and an excess consumption of sugar. Ludwig-Borycz et al. reported significantly lower adherence to the PHD among adolescents with lower socioeconomic status and educational attainment. Studies conducted in adult populations have similarly shown that those with greater adherence to the PHD were more likely to be female, white, and have a high income [18,19,20,21].

The relationship between adherence to the PHD and nutritional intake has also been investigated. Berthy et al. conducted a study in French adults and reported that increased adherence was associated with increased intakes of iron, calcium, and iodine, and a decreased intake of vitamin B12 [22]. However, Vallejo et al. found no significant difference in intakes of iron, calcium, iodine, vitamin B12, or zinc between differing levels of adherence to the PHD [17]. Both studies reported the positive association of the PHD with fibre intake, and the negative association with protein intake.

There is a need to understand how current diets may need to change to become healthier and more sustainable, and the PHD provides a reference point to assess this. As the PHD recommendations have not been specifically developed for adolescents, there is an additional need to assess whether these guidelines are suitable for meeting nutritional requirements in this population group. Adherence to the PHD, and the effect of PHD adherence on nutritional intake, have not been assessed in a UK adolescent population.

In this study, we aimed to explore adherence to the PHD guidelines in a UK adolescent population and assess the nutritional adequacy of the PHD in this population group. The specific research questions we explored were as follows: (1) to what extent do adolescents aged 11–15 years meet the PHD guidelines and how does this vary across sociodemographic groups? and (2) does adherence to the PHD provide adequate nutrient intake in adolescents aged 11–15 years?

## 2. Materials and Methods

### 2.1. Study Design and Setting

This study uses data collected as part of the UK Food provision, cUlture and Environment in secondary schooLs (FUEL) study; a cross-sectional study undertaken between 2019 and 2022 that evaluated the effects of school food policies in secondary schools [23]. For this, 24 h dietary intake data was collected from pupils aged 11–15 years in the Midlands region of England.

### 2.2. Study Population

We aimed to recruit a minimum of 34 schools from the Midlands, England using stratified sampling based on propensity score methods, as detailed in the study protocol [23]. A total of 482 eligible schools were identified and invited, with 36 schools participating. One class group each from the years 7 (age 11–12), 9 (age 13–14), and 10 (age 14–15) was selected, avoiding classes streamed by academic ability or optional subject classes. All pupils in selected classes were invited to participate with no exclusions; 2575 pupils were invited and 2543 consented. Participant information sheets were provided at least one week in advance to parents and pupils, and passive parental consent was gained via opt-out forms. Pupils provided written assent on the day of data collection.

### 2.3. Data Collection

The process for data collection and preparation and outcome variables is detailed in Figure 1. Participants completed an online 24 h dietary recall assessment using the Intake24 tool, and provided demographic information (including age, sex, ethnicity, and postcode) through a survey. Pupils self-reported their sex using the following categories: female, male, other, and would rather not say. Dietary data were collected on up to two occasions (school days only), 1–4 weeks apart. The first session was facilitated by researchers, and the second by teachers. Upon the completion of data collection, participating schools received a school-specific summary report and £300. Participating pupils received a £5 voucher.

The Intake24 tool uses the multiple pass method, which involves the uninterrupted recall of food intake before collecting detailed information for individual items [24]. The tool uses the NDNS food database to provide the food group and nutritional composition data of meals [25], and portion sizes were estimated using photographs. To reflect the study population, foods commonly consumed by minority ethnic groups were added to Intake24, and the adapted tool was piloted and revised prior to use in the study [26].

**Figure 1 nutrients-17-02140-f001:**
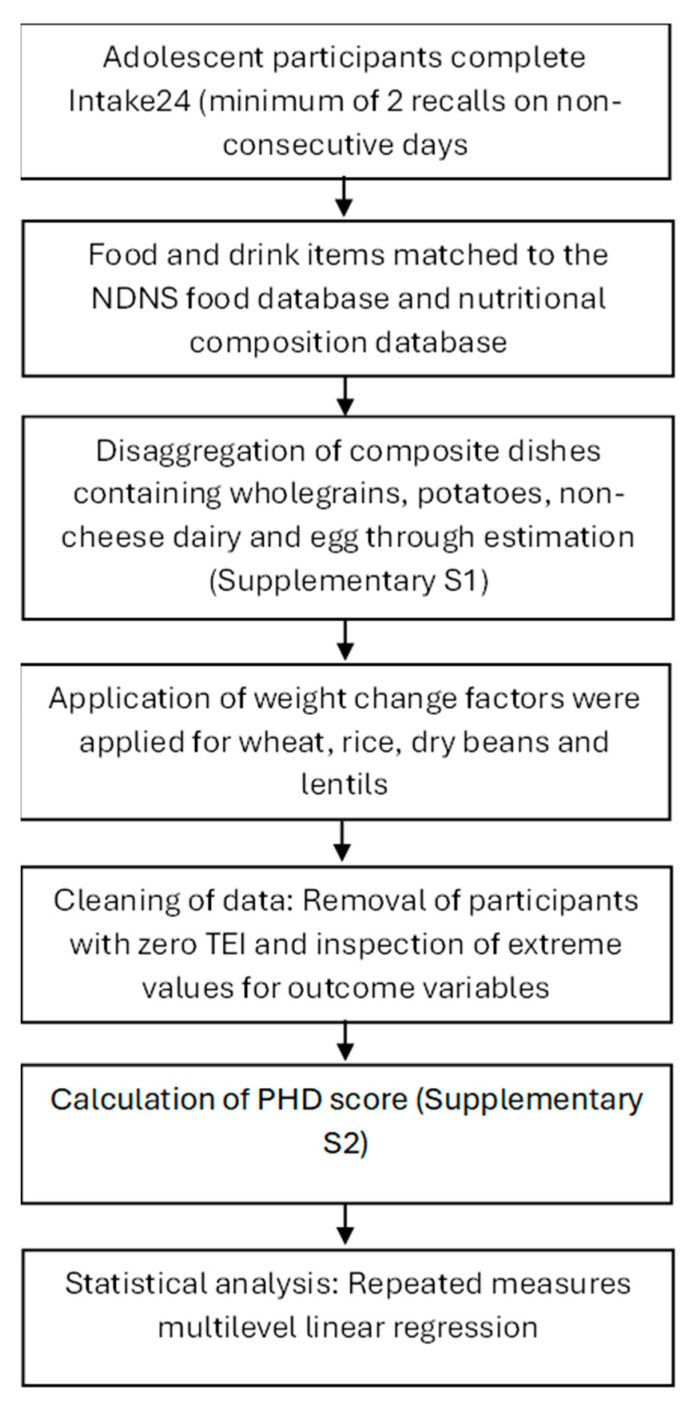
Process for data collection and preparation for outcome variables [25].

### 2.4. Data Cleaning

Participants with zero energy intake recorded were classed as having no dietary data, and were excluded for the purpose of this analysis, leaving 2273 participants remaining. Where age was missing (2.6% of sample), the average age from the participant’s class group was used. For sex, the ‘other’ and ‘would rather not say’ groups were combined. Postcodes were mapped to an Indices of multiple deprivation (IMD) quintile [27]. Where this was missing, the average IMD score for the participants in that school was used. Ethnicity was grouped into ‘White’, ‘Asian or Asian British’, ‘Black, African, Caribbean or Black British’, ‘Mixed or multiple’, ‘Other’, and ‘Prefer not to say’, and the latter two groups were combined due to small size.

### 2.5. Disaggregation of Composite Dishes

For each dietary recall, the Intake24 software estimated the consumption of all food groups relevant to the PHD except wholegrains, potatoes, non-cheese dairy, and egg. Therefore, the intake of these food groups was estimated manually.

Entries containing these food groups were identified by screening food codes. For items that were not considered ‘composite dishes’ (i.e., those composed of more than one food group), the full serving size was assigned to the relevant PHD group (e.g., the full weight of a portion of chips was assigned to ‘potato’). For composite dishes, a systematic approach to disaggregating components of the dish into their appropriate food groups was undertaken. This involved identifying the most commonly consumed dishes containing these food groups, and estimating the contribution of these dishes to the specified food groups using ingredients lists from supermarket websites, resulting in estimates for the intake of the food group in grams. Proportions were cross-checked with a food composition database from the Netherlands [28], as the required information was not available in the UK Composition of foods integrated dataset [25]. The use of ingredients lists for disaggregation has been demonstrated in the literature [29,30]. Full details of the disaggregation methods are available in Appendix A.

### 2.6. Preliminary Stages and Processing of Extreme Values

Weight change factors were applied for wheat, rice, dry beans, and lentils, as the PHD specifies raw weight values [31]. The 24 h intakes of each relevant food group and nutrient were summarised for each participant. Where participants had two 24 h dietary recall assessments, the mean intake was calculated. For all micronutrients, the percentage of the RNI met was calculated for each participant based on age and sex to allow for the meaningful interpretation of intake and comparison between groups. The percentage of recommended intake (RI) was calculated for fibre, and the percentage of total energy intake (TEI) was calculated for carbohydrates. The total intake value in grams was used for protein and omega-3 fats, as the protein intake guidelines are based on weight (which was not available in the FUEL study data), and there are no clear guidelines for omega-3 fat intake.

For each nutrient variable, intakes greater than three standard deviations (SDs) above the mean were identified, and the unprocessed Intake24 data was inspected for items contributing to the high intakes. Those that were due to an error in the Intake24 nutrient output were corrected. Where the same item had been entered multiple times for a participant (i.e., at the same mealtime and with the same portion size), this was assumed to be a participant or data processing error, and records for only one of these entries were used. A total of 45 records of food items were corrected due to erroneous output, and 5 were excluded due to repeated records.

### 2.7. Construction of a PHD Index Score

A PHD index based on the one developed by Stubbendorff et al. was used [32], with some adaptations (Table 1, see Appendix A). Food groups were classified as either ‘emphasised’ or ‘restricted’, and a score ranging from 0 to 3 was assigned to participants for each recommendation. Participants received more points for a higher consumption of emphasised foods and lower consumption of restricted foods. The maximum possible score was 39. For each recommendation, participants scoring 2 points or above were classified as having met the recommendation, as this generally corresponded with the lower or upper range of the relevant PHD recommendation, as appropriate.

### 2.8. Statistical Analysis

Statistical analysis was conducted using Stata/SE 18.0. The means and SDs were used to summarise continuous participant characteristics and PHD scores. The PHD scores were summarised overall and by sex, ethnicity, and IMD quintile. Absolute and percentage frequencies were used to summarise categorical participant characteristics and participants meeting each PHD recommendation. The percentage of participants meeting each recommendation was also summarised by sex, ethnicity, and IMD quintile, and a Chi-squared test was used to test for differences between groups. Statistical significance was determined using a two-sided *p*-value of <0.05.

Multilevel linear regression models were fitted with the PHD score as the explanatory variables and measures of nutrient intake as the outcome variables. Multilevel models were used to account for clustering at the school level; school ID was used as a random effect, and likelihood ratio (LR) tests were checked to ensure that this significantly improved the model fit. Residual vs. predicted value plots were used to ensure no heteroscedasticity, and histograms of the residual values were used to check the distribution.

Models were adjusted for age, sex, ethnicity, and IMD quintile. For the categorical variables sex, ethnicity, and IMD quintile, the baseline groups were female, white, and IMD quintile 1 (most deprived), respectively. Statistical significance was determined using a two-sided *p*-value of <0.05.

## 3. Results

### 3.1. Participant Characteristics

The participant characteristics are presented in Table 2. The mean age of participants was 13.7, and over half of the participants were female. White was the largest ethnic group (69.3% of the sample), followed by Asian or Asian British (15.8%). The highest proportion of participants was from the most deprived quintile (25.4%). Less than half (46.0%) of participants had two 24 h dietary recall records.

### 3.2. Adherence to the PHD

The mean total PHD index score was 14.2 (SD: 3.3) (Table 2). Higher scores were seen in females [14.6 (SD: 3.3)], the least deprived IMD quintile [14.5 (SD: 3.5)], and the Asian [14.6 (SD: 3.3)] and Other [14.6 (SD: 3.9)] ethnic groups.

Adherence to individual recommendations across participants is summarised in Figure 2. Adherence was highest for eggs and dairy (with 95.2% and 92.0% of participants meeting the recommendations, respectively) and lowest for unsaturated oils (0.2%), nuts (1.3%), legumes (5.5%), wholegrains (7.0%), vegetables (7.7%), and fish (14.8%).

More females than males met the recommendations for 9 out of 13 components (Table 3). A significant difference in adherence by sex was seen for four food groups; the highest adherence was seen in the ‘other’ group for potatoes (84.2%) and dairy (96.1%), and in females for beef, lamb, and pork (62.6%) and fish (16.7%).

A significant difference in adherence by ethnicity was seen for dairy (highest adherence in the Mixed group (94.5%)), beef, lamb, and pork (highest adherence in the Other group (73.6%)), chicken (highest adherence in the White group (73.4%)), and eggs (highest adherence in the Mixed group (96.9%)) (Table 4). For dairy and eggs, the Asian group had the lowest adherence (87.7% and 92.2%, respectively).

Adherence to recommendations was higher in IMD quintile 5 compared to IMD quintile 1 for 9 out of 13 components (Table 5). A significant difference in adherence by the IMD quintile was seen for potatoes and fruit (for which the highest adherence was seen in the least deprived group with 90% and 9.2% of this group meeting the recommendations, respectively), beef, lamb, and pork (where the highest adherence was seen in the most deprived group (61.8%)), and chicken (where the highest adherence was seen in the 2nd least deprived group (75.9%)).

### 3.3. Relationship Between PHD Index Score and Nutrient Adequacy

The intakes of all micronutrients decreased significantly with an increasing PHD index score, and remained significant following adjustment for age, sex, ethnicity, and IMD quintile (Table 6). The largest association was seen for vitamin B12, where intake decreased by an average of 12.9% (95% CI: −16.3, −9.4) of the RNI for every point increase in the PHD index score, and the smallest association was seen for vitamin D [regression co-efficient = −0.4% (95% CI: −0.7, −0.2) of RNI]. The decrease in iron, calcium, zinc, and iodine was approximately 2–3% of the RNI.

The association between the PHD index score and fibre intake was not significant. There was a small but significant increase for carbohydrates [regression co-efficient = 0.4% (95% CI: 0.3, 0.6) of TEI]. Omega-3 fat intake decreased by 0.04 g (95% CI: −0.06, −0.03) and protein by 2.8 g (95% CI: −3.2, −2.4).

## 4. Discussion

### 4.1. Adherence to Sustainable Diets

Adherence to the PHD was low, and no participants fully met the recommendations. Overall, adherence was much higher for restricted foods than emphasised foods. To meet PHD recommendations, adolescents need to substantially increase intakes of wholegrains, vegetables, fruit, fish, legumes, nuts, and unsaturated oils, and substantially decrease the consumption of sweeteners. A decrease in consumption of potatoes, chicken and red meat is also required. Intakes across sociodemographic groups were moderately consistent with the literature, with females and those living in the least deprived areas having the highest adherence to the PHD recommendations [18,19,20]. This study found Asian participants to be the most adherent, rather than white participants, which has been reported in other studies [18,33]. Furthermore, the higher adherence to the government’s ‘5-A-Day’ recommendation seen amongst white adults was not comparable with the results of this study [6], where there was no significant difference in adherence to fruit and vegetable guidelines between ethnic groups. Notably, participants from the most deprived areas in this study were the least likely to meet the recommendations for emphasised fruits.

Climate change and awareness of the need to adapt behaviours is high amongst adolescents, and 50% of those taking part in the InterClimate Network survey (which surveyed UK secondary school students on their views, behaviours, and motivations surrounding climate action) reported making environmentally friendly changes to their diet [34]. However, a scoping review exploring perceptions of sustainable diets amongst adolescents found that the understanding of sustainable diets was narrow and often conflated with healthy diets [35]. The complexity of sustainable diets was reported as a barrier to consumption, and an expert panel identified that this was a barrier particularly when health and sustainability were perceived to be conflicting [36]. Cost was identified as another key barrier—especially by those from lower socioeconomic backgrounds [35], and this may be demonstrated by the most deprived group having the lowest adherence to the PHD in this study.

Other factors dominating food choices included taste, and facilitators of consuming sustainable foods included simple dietary swaps or changes that required minimal cognitive choice, such as increasing plant-based options and avoiding labelling vegetarian options as ‘alternatives’ [35]. Easy-to-understand labelling, dietary education, and complementary ethical or moral values were also identified as facilitators.

### 4.2. Nutrient Adequacy of the PHD in Adolescent Populations

These findings of a negative association between adherence to the PHD and micronutrient intake are in contrast with those of Vallejo et al., where no significant differences in intakes of micronutrients were reported between differing levels of PHD adherence. [17]. These results are also in contrast with those of Berthy et al. for iron, calcium, and iodine, who observed a positive association [22]. However, our findings are consistent with the Berthy study for vitamin B12 and zinc, where a negative association was reported for vitamin B12, and higher zinc inadequacy was associated with higher adherence to the PHD. For macronutrients, the decrease in protein intake associated with higher PHD adherence was consistent with previously reported findings; however, the increase in fibre intake that has been previously reported was not observed in this study [17,22].

The inconsistent associations between PHD adherence and nutrient intakes seen across different studies may be explained by the varying patterns of adherence to the PHD recommendations. For example, in this study, adherence to high-fibre foods including fruit, vegetables, and wholegrains (emphasised foods) was low, which may provide a reason for why we did not find an association between PHD adherence and fibre intake. It is possible that participants with higher PHD scores have a lower consumption of restricted foods compared with those with lower scores, rather than having an increased consumption of emphasised foods. No participants were fully meeting the recommendations; therefore, it is not possible to assess whether full adherence to the PHD is associated with nutritional adequacy within the current sample.

Although this study has shown a decrease in some micronutrient intakes with increasing adherence to the PHD, those in the least deprived quintile and the Asian group had higher adherence to the PHD guidelines, and higher intakes of these micronutrients (data available on request). This may be due to the affordability of some healthy emphasised foods such as fruit and vegetables (a higher consumption of these were seen in less deprived groups), and the cultural variation in diets across ethnic groups.

### 4.3. Implications

The association between greater adherence to the PHD in adolescents’ current diets and meeting some nutrient recommendations is concerning, given that micronutrient intake is low in this population. Particular attention is needed to ensure that the promotion of adherence to the PHD guidelines does not disadvantage the most deprived individuals. An inadequate intake of micronutrients may lead to deficiencies. Inadequate calcium may have implications for conditions such as osteoporosis later in life [8]. Iron deficiency anaemia can cause clinical symptoms such as tiredness, and the risk of developing this is already high in adolescent girls due to menstrual blood loss [9]. Notably, adherence to PHD guidelines for restricted red meat (a reliable iron source) was lower in females. Similarly, inadequate vitamin B12 can lead to anaemia and neurological symptoms [37]. Adaptations to the PHD or supplementary guidance may be required to ensure that adolescents meet their nutritional requirements. This should include recommendations for foods high in the identified micronutrients within the emphasised food groups that will contribute to meeting the RNIs for this population group. Other measures may involve the fortification of foods or additional supplementation.

Provided that adequate nutrient intake can be achieved with the PHD for this population, the PHD should inherently remove the barrier of conflict or confusion between a healthy and sustainable diet identified by adolescents. However, action is required at multiple levels to enable adolescents to meet the PHD, including reducing the cost of emphasised foods, increasing the supply of tasty and appealing plant-based meals, and appropriate food labelling and education.

### 4.4. Strengths and Limitations

The strengths of this study include the high pupil participation rate and the diverse population, for which the data collection tool was adapted to capture dietary variation between ethnic groups. The use of stratified sampling supported the inclusion of schools with varied characteristics and a diverse pupil population, which was representative of the national population [38,39]. The multiple pass method used to recall dietary intake is the most accurate self-report method of dietary data collection in adolescents [40], and the Intake24 tool’s feature to use prompt questions further increased accuracy. The use of a graded PHD score allowed for a more nuanced assessment of the association between nutritional intake and PHD adherence, compared to using a binary scoring system. Nutrient intakes were expressed as a percentage of the relevant recommendation, allowing for the assessment of nutrient adequacy and comparison between nutrients. Finally, the use of multilevel models accounted for clustering at the school level.

There were some study limitations. Participating schools may have differed to non-participating schools in characteristics not captured in the sampling approach, e.g., quality of food provision, impacting the generalizability of the findings. Although a systematic approach was used to disaggregate certain foods, there were disadvantages to this approach. For example, the proportions of foods in retail products on supermarket websites may differ from those in equivalent home-cooked meals, and some similar dishes were grouped in the current study, which may have resulted in some misestimation. Half of the participants only had one dietary record, so habitual dietary intake is unlikely to have been captured, particularly given that PHD recommendations are better interpreted over a longer period of time. For example, the recommendation of 25 g of egg per day can be interpreted as two eggs a week. As the data collection was supervised, this may have introduced a social acceptability bias, and the use of self-report dietary assessment may have resulted in misreporting, which may be a particular issue in this age group [41]. The PHD score still simplifies the complexity of an overall diet, particularly for foods where some, but not excess, consumption can be beneficial. As no participants fully adhered to the PHD, it was not possible to explore how full adherence was associated with nutritional adequacy.

### 4.5. Implications for Future Research

Further research is required to examine the effect of the PHD on nutrient intake in adolescents using a measure of usual dietary intake. Analysis in a participant sample where the full PHD recommendations have been met by some would allow for a better assessment of the nutritional adequacy of the PHD in the adolescent population. Additionally, research into the effect of adherence to individual recommendations will be beneficial, as currently it is unclear if the negative association between PHD adherence and micronutrient intake observed in this sample is due to the low overall consumption of some recommended foods. Long-term follow-up for health outcomes is also needed.

There is a need for a global PHD index score, as variation in the methodology between studies makes comparisons between populations difficult. Further research may also inform adaptations to the PHD for adolescents, or additional guidelines for vitamin supplementation or the fortification of foods if the PHD is followed.

## 5. Conclusions

There is a need for individuals to change their dietary behaviour and move towards a PHD to lessen the detrimental impact of the food system on the planet, but this study suggests that there still needs to be substantial behavioural shifts in the adolescent age group. This will require action at multiple levels to support adolescents in this change. However, questions regarding the nutritional adequacy of the PHD based on current typical dietary patterns in this population must first be further understood and addressed. This should be followed by the review and potential adaptation of the PHD for an adolescent population, considering current dietary patterns and nutritional requirements for this age group.

## Figures and Tables

**Figure 2 nutrients-17-02140-f002:**
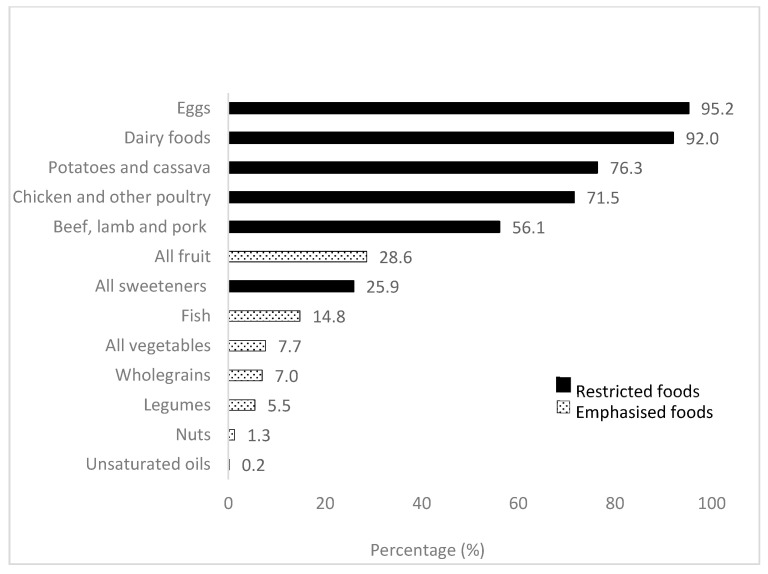
Percentage of participants meeting each PHD recommendation.

**Table 1 nutrients-17-02140-t001:** Constructed PHD index, adapted from [32].

	EAT-Lancet Recommendations (g/day)	3 Points	2 Points	1 Point	0 Points
Wholegrains *	232 (0–60% of energy)	>232	116–232	58–116	<58
Potatoes and cassava	50 (0–100)	<50	50–100	100–150	>150
All vegetables *	300 (200–600)	>300	200–300	100–200	<100
All fruit *	200 (100–300)	>200	100–200	50–100	<50
Dairy foods	250 (0–500)	<250	250–500	500–750	>750
Beef, lamb, and pork	14 (0–28)	<14	14–28	28–42	>42
Chicken and other poultry	29 (0–58)	<29	29–58	58–87	>87
Eggs	13 (0–25)	<13	13–25	25–37.5	>37.5
Fish *	28 (0–100)	>28	14–28	7–14	<7
Dry beans, lentils, and peas *^,†^	50 (0–100)	>75	37.5–75	18.75–37.5	<18.75
Soy foods *^,†^	25 (0–50)				
Peanuts *^,‡^	25 (0–75)	>50	25–50	12.5–25	<12.5
Tree nuts *^,‡^	25				
Unsaturated oils	40 (20–80)	>40	20–40	10–20	<10
All sweeteners	31 (0–31)	<15.5	15.5–31	31–46.5	>46.5

* Emphasised foods. ^†^ Combined to form group ‘legumes’. ^‡^ Combined to form group ‘nuts’.

**Table 2 nutrients-17-02140-t002:** Characteristics of participants included in analysis and PHD score by sex, ethnic group, and IMD quintile.

	Mean (SD)	Number (%)	PHD Score, Mean (SD)
Total		2273 (100)	14.2 (3.3)
Age, years	13.7		
**Sex**			
Female		1269 (55.8)	14.6 (3.3)
Male		928 (40.8)	13.6 (3.2)
Other *		76 (3.3)	14.8 (3.3)
**Ethnic group**			
White		1576 (69.3)	14.1 (3.3)
Asian or Asian British		359 (15.8)	14.6 (3.3)
Black, African, Caribbean, or Black British		123 (5.4)	13.9 (3.4)
Mixed or multiple		128 (5.6)	14.4 (3.3)
Other ^†^		87 (3.8)	14.6 (3.9)
**IMD quintile**			
1 (most deprived)		578 (25.4)	13.8 (3.2)
2		379 (16.7)	14.1 (3.4)
3		457 (20.1)	14.4 (3.2)
4		444 (19.5)	14.3 (3.3)
5 (least deprived)		415 (18.3)	14.5 (3.5)
**Number of 24 h recalls**			
1		1227 (54.0)	
2		1046 (46.0)	

* ’Other’ and ‘would rather not say’ groups combined. ^†^ ‘Other’ and ‘prefer not to say’ groups combined.

**Table 3 nutrients-17-02140-t003:** Percentage of participants meeting PHD recommendations by sex.

	Number (%) Meeting PHD Recommendation			
	Female	Male	Other	*p*-Value
Wholegrains	80 (6.3)	75 (8.1)	5 (6.6)	0.27
Potatoes and cassava	945 (74.5)	726 (78.2)	64 (84.2)	0.03 *
All vegetables	100 (7.9)	69 (7.4)	5 (6.6)	0.87
All fruit	385 (30.3)	248 (26.7)	16 (21.1)	0.06
Dairy foods	1202 (94.7)	815 (87.8)	73 (96.1)	<0.001 *
Beef, lamb, and pork	794 (62.6)	439 (47.3)	41 (54.0)	<0.001 *
Chicken and other poultry	914 (72.0)	649 (69.9)	62 (81.6)	0.08
Eggs	1207 (95.1)	883 (95.2)	74 (97.4)	0.67
Fish	212 (16.7)	114 (12.3)	11 (14.5)	0.02 *
Legumes	77 (6.1)	43 (4.6)	5 (6.6)	0.32
Nuts	13 (1.0)	16 (1.7)	0 (0.0)	0.21
Unsaturated oils	3 (0.2)	1 (0.1)	0 (0.0)	0.73
All sweeteners	333 (26.2)	228 (24.6)	28 (36.8)	0.06

* Significant *p*-values.

**Table 4 nutrients-17-02140-t004:** Percentage of participants meeting PHD recommendations by ethnicity.

	Number (%) Meeting PHD Recommendation					
	White	Asian	Black	Mixed	Other	*p*-Value
Wholegrains	104 (6.6)	31 (8.6)	8 (6.5)	13 (10.2)	4 (4.6)	0.33
Potatoes and cassava	1198 (76.0)	269 (74.9)	96 (78.1)	102 (79.7)	70 (80.5)	0.68
All vegetables	114 (7.2)	26 (7.2)	15 (12.2)	12 (9.4)	7 (8.1)	0.33
All fruit	472 (30.0)	88 (24.5)	27 (22.0)	35 (27.3)	27 (31.0)	0.12
Dairy foods	1461 (92.7)	315 (87.7)	113 (91.9)	121 (94.5)	80 (92.0)	0.03 *
Beef, lamb, and pork	793 (50.3)	259 (72.1)	81 (65.9)	77 (60.2)	64 (73.6)	<0.001 *
Chicken and other poultry	1157 (73.4)	250 (69.6)	77 (62.6)	80 (62.5)	61 (70.1)	0.010 *
Eggs	1510 (95.8)	331 (92.2)	115 (93.5)	124 (96.9)	84 (96.6)	0.04 *
Fish	240 (15.2)	59 (16.4)	11 (8.9)	17 (13.3)	10 (11.5)	0.26
Legumes	89 (5.7)	25 (7.0)	2 (1.6)	5 (3.9)	4 (4.6)	0.21
Nuts	17 (1.1)	6 (1.7)	2 (1.6)	1 (0.8)	3 (3.5)	0.34
Unsaturated oils	2 (0.1)	2 (0.6)	0 (0.0)	0 (0.0)	0 (0.0)	0.44
All sweeteners	396 (25.1)	108 (30.1)	27 (22.0)	37 (28.9)	21 (24.1)	0.24

* Significant *p*-values.

**Table 5 nutrients-17-02140-t005:** Percentage of participants meeting PHD recommendations by IMD quintile.

	Number (%) Meeting PHD Recommendation					
	IMD Quintile 1 (Most Deprived)	IMD Quintile 2	IMD Quintile 3	IMD Quintile 4	IMD Quintile 5 (Least Deprived)	*p*-Value
Wholegrains	29 (5.0)	31 (8.2)	32 (7.0)	35 (7.9)	33 (8.0)	0.25
Potatoes and cassava	404 (69.9)	275 (72.6)	367 (80.3)	353 (79.5)	336 (90.0)	<0.001 *
All vegetables	33 (5.7)	27 (7.1)	31 (6.8)	45 (10.1)	38 (9.2)	0.06
All fruit	129 (22.3)	108 (28.5)	148 (32.4)	123 (27.7)	141 (34.0)	<0.001 *
Dairy foods	537 (92.9)	353 (93.1)	426 (93.2)	406 (91.4)	368 (88.7)	0.072
Beef, lamb, and pork	357 (61.8)	218 (57.5)	268 (58.6)	226 (50.9)	205 (49.4)	<0.001 *
Chicken and other poultry	388 (67.1)	275 (72.6)	321 (70.2)	337 (75.9)	304 (73.3)	0.029 *
Eggs	556 (96.2)	364 (96.0)	436 (95.4)	419 (94.4)	389 (93.7)	0.34
Fish	82 (14.2)	48 (12.7)	60 (13.1)	71 (16.0)	76 (18.3)	0.13
Legumes	32 (5.5)	25 (6.6)	21 (4.6)	25 (5.6)	22 (5.3)	0.80
Nuts	3 (0.5)	6 (1.6)	5 (1.1)	9 (2.0)	6 (1.5)	0.28
Unsaturated oils	1 (0.2)	0 (0.0)	1 (0.2)	0 (0.0)	2 (0.5)	0.45
All sweeteners	144 (24.9)	99 (26.1)	130 (28.5)	98 (22.1)	118 (28.4)	0.16

* Significant *p*-values.

**Table 6 nutrients-17-02140-t006:** Association between PHD index score and nutrient intake, unadjusted and adjusted models.

		Unadjusted		Adjusted ^†^	
		Regression Co-Efficient (CI)	*p*-Value	Regression Co-Efficient (CI)	*p*-Value
	**Micronutrients**				
% RNI	Iron	−2.7 (−3.3, −2.1)	<0.001	−2.1 (−2.7, −1.5)	<0.001 *
	Calcium	−3.0 (−3.8, −2.1)	<0.001	−3.2 (−4.1, −2.3)	<0.001 *
	Vitamin B12	−13.8 (−17.2, −10.4)	<0.001	−12.9 (−16.3, −9.4)	<0.001 *
	Vitamin D	−0.5 (−0.8, −0.2)	0.001	−0.4 (−0.7, −0.2)	0.003 *
	Zinc	−3.3 (−3.8, −2.8)	<0.001	−3.2 (−3.7, −2.7)	<0.001 *
	Iodine	−2.4 (−3.1, −1.6)	<0.001	−2.2 (−2.9, −1.4)	<0.001 *
	**Macronutrients**				
% RI	Fibre	−0.1 (−0.5, 0.3)	0.70	0.01 (−0.4, 0.4)	0.97
% TEI	Carbohydrates	0.4 (0.3, 0.5)	<0.001	0.4 (0.3, 0.6)	<0.001 *
g	Omega−3 fats	−0.04 (−0.06, −0.03)	<0.001	−0.04 (−0.06, −0.03)	<0.001 *
	Protein	−2.9 (−3.3, −2.6)	<0.001	−2.8 (−3.2, −2.4)	<0.001 *

* Significant *p*-values. ^†^ Adjusted for age, sex, ethnicity, and IMD quintile.

## Data Availability

The raw data supporting the conclusions of this article will be made available by the authors on request.

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
