# Peer review of "Sustainability of Diets Consumed by UK Adolescents and Associations Between Diet Sustainability and Meeting Nutritional Requirements"

_nutrients, 2025, doi:10.3390/nu17132140_

Round 1
Reviewer 1 Report
Comments and Suggestions for Authors
This is an interesting and relevant work performed by Ashraf et al on the sustainability of diets consumed by UK adolescents and associations between diet sustainability and meeting nutritional requirements. I believe it can be considered for publication after the following revisions be done:
Abstract: What can be done next, based on the obtained results in this study? Include some information about this at the end of the Conclusions.
Introduction: Some relevant studies conducted in other regions outside UK should be included.
Considering the high number of tasks and subsections in the Materials and Methods, I suggest the inclusion of a flowchart for a better visualization of all these steps you took to perform your research.
Line 215 – Revise this: “Participant characteristics are presented in Error! Reference source not found.. The”.
The Results and Discussion sections are properly described and I have no further comments.
Similarly to the abstract, future perspectives should be included in the end of the conclusions.
Author Response
Thank you for your review, this is much appreciated. Please see attached for our response to your review and the edited manuscript with changes indicated by red text.

Reviewer 2 Report
Comments and Suggestions for Authors
The authors conducted a cross-sectional study to investigate th sustainability of diets consumed by UK adolescents and associations between diet sustainability and meeting nutritional requirements. Overall the manuscript is well-written and this is an interesting study. However, I have a few comments that I think could help strengthen the presentation of the methods and results.
- samples are collected from 36 out of 482 schools. This seems to be a large attrition rate. Any characteristics that are different between the schools agreed and those that did not agree? Any bias that could be introduced by this attrition rate? Are the findings from these 36 schools generalizable to all the schools?
- any clustered effects between the schools? or between the school years?
- any potential recall bias from the dietary questionnaire?
- in line 203, please indicate that pvalue is two sided or one sided.
< !-- notionvc: a5955d87-b044-4cba-94b2-5f073715b8c7 -->
Author Response

(The authors gave the same response as above.)
